# StoryGPT-V: Large Language Models as Consistent Story Visualizers

## Abstract

Recent generative models have demonstrated impressive capabilities in generating realistic and visually pleasing images grounded on textual prompts. Nevertheless, a significant challenge remains in applying these models for the more intricate task of story visualization. Since it requires resolving pronouns (he, she, they) in the frame descriptions, i.e., anaphora resolution, and ensuring consistent characters and background synthesis across frames. Yet, the emerging Large Language Model (LLM) showcases robust reasoning abilities to navigate through ambiguous references and process extensive sequences. Therefore, we introduce *StoryGPT-V*, which leverages the merits of the latent diffusion (LDM) and LLM to produce images with consistent and high-quality characters grounded on given story descriptions. First, we train a character-aware LDM, which takes character-augmented semantic embedding as input and includes the supervision of the cross-attention map using character segmentation masks, aiming to enhance character generation accuracy and faithfulness. In the second stage, we enable an alignment between the output of LLM and the character-augmented embedding residing in the input space of the first-stage model. This harnesses the reasoning ability of LLM to address ambiguous references and the comprehension capability to memorize the context. We conduct comprehensive experiments on two visual story visualization benchmarks. Our model reports superior quantitative results and consistently generates accurate characters of remarkable quality with low memory consumption. Our code will be made publicly available[1].

## 1 Introduction

Image generation algorithms have made significant strides and are on the verge of matching human-level proficiency. Despite this progress, even a powerful image generator suffers from story visualization task, which involves generating a series of frames that maintain semantic coherence based on narrative descriptions (Li et al., 2019; Zeng et al., 2019; Maharana et al., 2021; 2022). This challenge arises from the fact that captions for a single image are typically self-sufficient, lacking the continuity needed to capture the narrative of object interactions that unfold through multiple sentences over a sequence of frames. This poses a promising avenue for further research and exploration of story visualization. Such a task demands a model capable of producing high-quality characters and detailed environmental objects grounded on given text descriptions. Moreover, it requires the ability to disambiguate referential pronouns in the subsequent frame descriptions, e.g., "*she, he, they*".

Prior studies (Maharana & Bansal, 2021; Li et al., 2019; Maharana et al., 2022; Song et al., 2020b; Chen et al., 2022a) explore the realm of story visualization but do not take reference resolution (Seo et al., 2017) (i.e., anaphora resolution in the context of natural language processing (Aone & William, 1995; McCarthy & Lehnert, 1995)) into consideration. Story-LDM (Rahman et al., 2023) first extended story visualization benchmarks with referential text and devises an attention memory module that retains visual context throughout the series of generated frames. However, it still struggles to generate precise characters for referential text since the interaction between current descriptions and contextual information occurs within the CLIP (Radford et al., 2021b) semantic space, causing a loss in fine-grained language understanding and hindering referencing capabilities. Furthermore, the attention memory module requires maintaining all previous images in latent

---

[1]Please refer to the anonymous webpage for qualitative results.

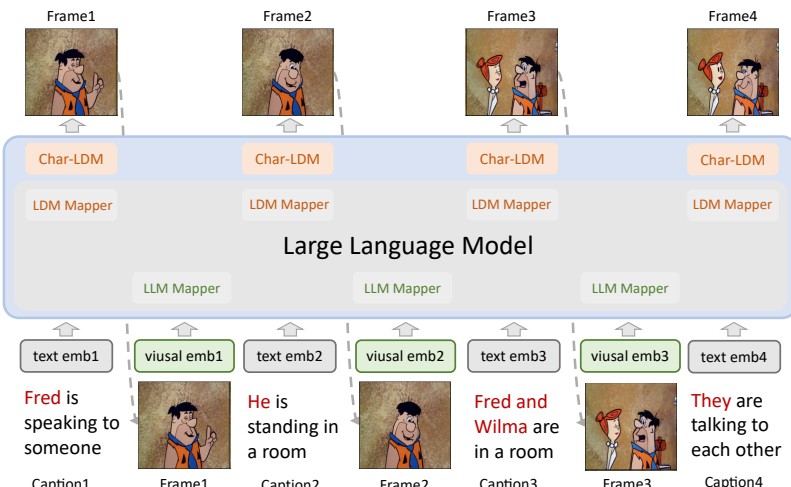

Figure 1: We present StoryGPT-V, which empowers a large language model for interleaved image-text comprehension and aligns its output with character-aware Latent Diffusion (Char-LDM) for autoregressive story visualization grounded on co-referential text descriptions.

pixel space for attention calculations, significantly increasing memory demands with each additional frame in autoregressive generation.

The limitation of previous works leads us to rethink how to achieve accurate and efficient reference resolution toward consistent story visualization. Large Language Models (LLMs) (Radford et al., 2019; Raffel et al., 2020; Brown et al., 2020; Zhang et al., 2022), trained on extensive text corpora, have exhibited impressive capabilities in deciphering contextual references in natural language descriptions. Prior works (Koh et al., 2023a; Ge et al., 2023) have demonstrated the effectiveness of harnessing LLMs for tasks involving image comprehension and generation, where the visual features are adapted within LLM's token space rather than the pixel space. Hence, such a model could be utilized to efficiently address ambiguous references for story visualization tasks.

In this work, we aim at story visualization grounded on given co-referential frame descriptions, focusing on delivering high-quality and coherent portrayals of characters. To achieve this, we leverage a powerful text-to-image model (Rombach et al., 2022) to generate high-quality characters and environmental objects grounded on given frame descriptions, coupled with the reasoning ability of Large Language Models (LLMs) to resolve ambiguous references and improve the cohesiveness of the context. To improve the generation of highly faithful characters, we enhance the pre-trained Latent Diffusion (LDM) towards character-aware training in the first stage. We first augment the token feature by incorporating the visual representation of the corresponding character. Additionally, we regulate the cross-attention map of the character token to highlight the interaction between the conditional token and specific latent pixels.

In addressing the challenge of ambiguous reference, which cannot be effectively handled by a robust text-to-image model alone, we leverage an LLM that takes interleaved images and co-referential frame descriptions as input, and aligns its visual output with the character-augmented embedding encoded by first-stage model. Such semantic guidance, along with LLM's casual modeling, enables effective reference resolution and consistent generation. Furthermore, our approach efficiently preserves context by processing images as sequences of tokens in the LLM input space with low memory consumption.

**Contributions.** Our contributions are as follows:

- We enhance the text representation by integrating the visual features of the corresponding characters, then refine a character-aware LDM for better character generation by directing cross-attention maps with character segmentation mask guidance.

- We adapt LLM by interlacing text and image inputs, empowering it to implicitly deduce references from previous contexts and produce visual responses that align with the input space of the first-stage Char-LDM. This leverages the LLM's reasoning capacity for reference resolution and the synthesis of coherent characters and scenes.

- Our model is capable of visualizing stories featuring precise and coherent characters and backgrounds on story visualization benchmarks. Furthermore, we showcase the model's proficiency in producing extensive (longer than 40 frames) visual stories with low memory consumption.

## 2 RELATED WORK

**Text-to-image synthesis.** Numerous works (Crowson et al., 2022; Gafni et al., 2022; Ding et al., 2021) have demonstrated unprecedented performance on semantic generation. Recently, diffusion-based text-to-image models (Ramesh et al., 2022; 2021; Rombach et al., 2022; Saharia et al., 2022) have shown significant advancements in enhancing image quality and diversity through the utilization of diffusion models. However, these text-to-image approaches primarily concentrate on aligning individual-generated images grounded on text descriptions and do not take into account the crucial aspects of character and scene consistency across multiple frames in the story visualization task. Additionally, they lack the capability to effectively resolve co-reference issues within a narrative description.

**Multi-modal Large Language Models.** Large Language Models (LLMs) wield an extensive repository of human knowledge and exhibit impressive reasoning capabilities. Recent studies (Tsimpoukelli et al., 2021; Chen et al., 2022b; Alayrac et al., 2022; Li et al., 2023b) utilize pre-trained language models to tackle vision-language tasks, and subsequent studies (Zhu et al., 2023; Zhang et al., 2023; Wang et al., 2023; Li et al., 2023a; Huang et al., 2023; Chen et al., 2023) further enhance multi-modal abilities by aligning vision models with LLM input space. In addition to multi-modal comprehension, several works are dedicated to more challenging multi-modal generation tasks. FROMAGe (Koh et al., 2023b) appends a special retrieval token to LLM and maps the hidden representation of this token into a vector space for retrieving images. Several current works (Koh et al., 2023a; Wu et al., 2023; Zeqiang et al., 2023) learn a mapping from hidden embeddings of an LLM represents for additional visual outputs into the input space of a frozen pre-trained text-to-image generation model (Rombach et al., 2022). In this work, we fed multi-modal LLM with interleaved image and referential text descriptions as input and aligned the output with a character-aware fused embedding from our first-stage Char-LDM, guiding the LLM in implicitly deducing the references.

**Story Visualization.** StoryGAN (Li et al., 2019) pioneers the story generation task, which proposes a sequential conditional GAN framework with dual frame and story level discriminators to improve image quality and narrative coherence. DuCoStoryGAN (Maharana et al., 2021) introduces a dual-learning framework that utilizes video captioning to enhance semantic alignment between descriptions and generated images. VLCStoryGAN (Maharana & Bansal, 2021) used video captioning for semantic alignment between text and frames. Recently, StoryDALL-E (Maharana et al., 2022) retrofits the cross-attention layers of the pre-trained text-to-image model to promote generalizability to unseen visual attributes of the generated story. These methods do not consider ambiguous references in text descriptions. Story-LDM (Rahman et al., 2023) first introduced reference resolution in story visualization tasks and proposed an autoregressive diffusion-based framework with a memory-attention module to resolve ambiguous references. Nevertheless, it struggled with accurately resolving references and was memory-intensive, as it required retaining all previous context in pixel space. In our work, we employ a powerful causal inference LLM for reference resolution, and it efficiently maintains context by mapping visual features into several token embeddings as LLM inputs rather than operating in latent pixel space.

## 3 METHODS

The objective of story visualization is to transform a textual narrative, composed of a series of $N$ descriptions $S_1, ...S_N$, into a sequence of corresponding visual frames $I_1, ..., I_N$ that illustrate the story. We've developed a two-stage method aimed at generating temporally consistent visual stories

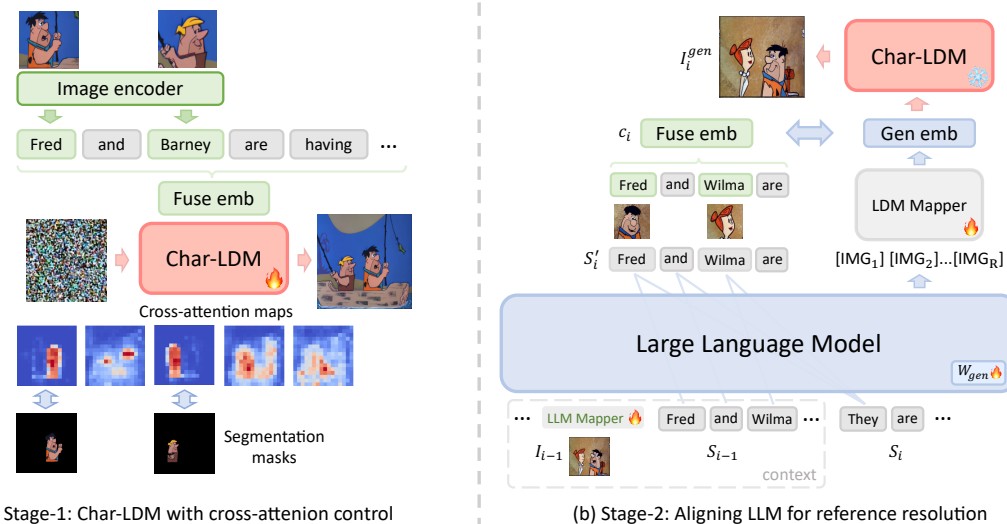

(a) Stage-1: Char-LDM with cross-attention control      (b) Stage-2: Aligning LLM for reference resolution

Figure 2: (a) In the first stage, a fused embedding is created by integrating character visuals with text embeddings, serving as the Char-LDM's conditional input, and the cross-attention maps of Char-LDM will be guided by corresponding character segmentation mask for accurate and high-quality character generation (section 3.2). (b) In the second stage, the LLM takes the interleaved image and text context as input and generates $R$ [IMG] tokens. These tokens are then projected by LDM Mapper into an intermediate output, which will be encouraged to align with fused embedding as Char-LDM's input. The figure intuitively shows how the character-augmented fused embedding and the casual language modeling aid LLM for reference resolution (section 3.3).

with accurate and high-quality characters. First, we augment text representation with characters' visual features and refine a character-aware LDM (Rombach et al., 2022) (Char-LDM) towards high-quality character generation. This is achieved by directing the cross-attention maps of specific tokens associated with the corresponding characters, using character segmentation mask supervision (section 3.2). Then, we leverage the reasoning ability of LLM to resolve ambiguous references by aligning the output of LLM with Char-LDM input space for temporal consistent story visualization (section 3.3).

## 3.1 PRELIMINARIES

**Cross-attention in text-conditioned Diffusion Models.** In diffusion models (Ho et al., 2020; Song et al., 2020a), each diffusion step $t$ involves predicting noise $\epsilon$ from the noise code $z_t \in \mathbb{R}^{(h \times w) \times d_v}$ conditioned on text embedding $\psi(S) \in \mathbb{R}^{L \times d_c}$ via U-shaped Network (Ronneberger et al., 2015), where $\psi$ is the text encoder, $h$ and $w$ are the latent spatial dimensions and $L$ is the sequence length. Within U-Net, the cross-attention layer accepts the spatial latent code $z$ and the text embeddings $\psi(S)$ as inputs, then projects them into $Q = W^q z$, $K = W^k \psi(S)$ and $V = W^v \psi(S)$, where $W^q \in \mathbb{R}^{d_v \times d'}$, $W^k, W^v \in \mathbb{R}^{d_c \times d'}$. The attention scores is computed as $A = \text{Softmax}(\frac{QK^T}{\sqrt{d'}}) \in \mathbb{R}^{(h \times w) \times L}$, where $A[i, j, k]$ represents the attention of $k$-th text token to the $(i, j)$ latent pixel. In this context, each entry $A[i, j, k]$ within the cross-attention map $A$ quantifies the magnitude of information propagation from the $k$-th text token to the latent pixel at position $(i, j)$. This feature of the interaction between semantic representation and latent pixels is harnessed in various tasks such as image editing (Hertz et al., 2022; Parmar et al., 2023), video editing (Liu et al., 2023b), and fast adaptation (Shi et al., 2023; Xiao et al., 2023; Couairon et al., 2023; Wei et al., 2023).

## 3.2 CHARACTER-AWARE LDM WITH ATTENTION CONTROL

**Integrate visual features with text conditions.** To achieve accurate and high-quality characters in story visualization, we augment text descriptions with visual features of corresponding characters and guide the attention of text conditions to focus more on corresponding character synthesis. Given a text description $S$, suppose there are $K$ characters that should be generated in image $I$, images of those characters $\{I_c^1, ..., I_c^K\}$, a list of token indices indicating each character name located in the description, denoted as $\{i_c^1, ...i_c^K\}$. Inspired by (Wei et al., 2023; Xiao et al., 2023; Ma et al., 2023), we first utilize CLIP (Radford et al., 2021b) text encoder $\psi$ and image encoder $\phi$ to obtain text embedding and visual features of the characters appear in the image respectively. Then, we augment the text embedding if the token represents a character name. More specifically, we concatenate the token embedding and the visual features of the corresponding character and feed them into an MLP to obtain the augmented text embedding. Each augmented token embedding in the augmented embedding $c$ is formulated as below:

$$c^k = \text{MLP}\left(\text{concat}\left((\psi(S[i_c^k]), \phi(I_c^k))\right)\right) \tag{1}$$

where $i_c^k$ refers to the index of the text token for character $k$, and $I_c^k$ the image corresponding to character $k$. The embeddings for tokens in $c$ that are unrelated to the character remain identical to the vanilla CLIP token embeddings. The enhanced embedding $c$ is then employed as supervision for the second-stage training, which will be further detailed in section 3.3, where $c_1, ...c_N$ are the corresponding augmented embeddings for $S_1, ..., S_N$.

**Controlling attention of text tokens.** Previous work (Hertz et al., 2022) has demonstrated that the visual characteristics of generated images are influenced by the intricate interplay between latent pixels and text embedding through the diffusion process of LDM (Rombach et al., 2022). However, in vanilla LDM (Rombach et al., 2022), a single latent pixel can unrestrictedly engage with all text tokens. Therefore, we introduce a constraint to refine this behavior and strengthen the impact of the token representing the character's name on certain pixels in the denoising process, as illustrated in fig. 2 (a). First, we obtain offline segmentation masks of corresponding characters denoted as $\{M_1, ...M_K\}$ as supervision signals via SAM (Kirillov et al., 2023). We then encourage the cross-attention map $A_k$ for each character $k$ at the token index position $i_c^k$, to align with the binary segmentation mask $M_k$, whereas diverging from irrelevant regions $\bar{M}_k$, formulated as follows:

$$\mathcal{L}_{\text{reg}} = \frac{1}{K} \sum_{k=1}^{K} (A_k^- - A_k^+) \tag{2}$$

where

$$A_k^- = \frac{A_k \odot \bar{M}_k}{\sum_{i,j}(\bar{M}_k)_{ij}}, \ A_k^+ = \frac{A_k \odot M_k}{\sum_{i,j}(M_k)_{ij}} \tag{3}$$

where $K$ is the number of characters to be generated in the image, $i_c^k$ is the index of text token representing character $k$ and $\odot$ is the Hadamard product. By reducing the loss, it increases the attention of character tokens to the relevant pixels of their respective characters, while reducing their attention to irrelevant areas. Moreover, as the token embeddings are enriched with the visual features of the corresponding character, this attention control serves to deepen the connection between the augmented semantic space and latent pixel denoising, which can consequently enhance the quality of synthesized characters.

Our first stage Char-LDM focuses solely on the quality of image generation grounded on a single caption. Yet, there remain challenges that surpass the abilities of text-to-image generators in visualizing a sequence of stories. Firstly, story visualization demands character and background consistency, an aspect not covered by our first-stage enhancements. Moreover, the inherent nature of lengthy descriptions includes referential terms like he, she, or they, which presents a significant challenge for LDM in achieving accurate inference. In contrast, LLMs can adeptly infer the intended character to which the ambiguous text refers. Therefore, to address this issue, we harness the formidable reasoning capabilities of LLM to disambiguate such references.

## 3.3 ALIGNING LLM FOR REFERENCE RESOLUTION

To enable an LLM to autoregressively generate images conditioned on prior context and resolve ambiguous references, the model must be capable of (i) processing images; (ii) producing images;

and (iii) implicitly deducing the subject of reference. The model could understand the image by learning a linear mapping from the visual feature to the LLM input space, and generate images by aligning the hidden states with conditional input required by LDM, which is the fused embedding encoded by first-stage Char-LDM's text and visual encoder. It integrates the visual features of characters into the text embedding. This character-augmented embedding, along with the causal language modeling (CLM) (Vaswani et al., 2017; Radford et al., 2018; 2019) will direct the LLM to implicitly deduce and generate the correct character for the referential input, as depicted in fig. 2 (b).

More specifically, the LLM input consists of interleaved co-referential text descriptions and story frames with flexible frame length $n$, in the order of $(I_1, S_1, ..., I_{n-1}, S_{n-1}, S_n)$, where $2 \leq n \leq N$. We first extract visual embeddings $\phi(I_i) \in \mathbb{R}^{d_i}$ with CLIP (Radford et al., 2021b) visual backbone, where $i \in [2, n]$, and learn $\texttt{Mapper}_{LLM}$ with trainable matrix $\mathbf{W}_{v2t} \in \mathbb{R}^{d_i \times me}$ which maps $\phi(I_i)$ into $m$ $k$-dimensional embeddings reside within LLM input space (Li et al., 2023b; Liu et al., 2023a; Zhu et al., 2023), where $e$ is the dimension of LLM embedding space. Additionally, like recent works (Koh et al., 2023a; Wu et al., 2023; Zeqiang et al., 2023) in enabling LLM to generate images, we add additional $R$ tokens, denoted as $\texttt{[IMG}_1\texttt{]}$, ..., $\texttt{[IMG}_R\texttt{]}$ to represent visual outputs and incorporate trainable matrix $\mathbf{W}_{gen} \in \mathbb{R}^{R \times e}$ into frozen LLM. The training objective is to minimize the negative log-likelihood of producing $\texttt{[IMG]}$ tokens conditioned on previously interleaved image/text tokens $\mathcal{T}_{prev}$:

$$\mathcal{L}_{gen} = -\sum_{r=1}^{R} \log p(\texttt{[IMG}_r\texttt{]}|\mathcal{T}_{prev}, \texttt{[IMG}_{<r}\texttt{]}) \tag{4}$$

where

$$\mathcal{T}_{prev} = \{\phi(I_{<i})^T \mathbf{W}_{v2t}, \psi(S_{1:i})\} \tag{5}$$

where $i \in [2, n]$ is the number of text descriptions of the current step. To align $\texttt{[IMG]}$ produced by LLM with LDM input space, we utilize a Transformer-based $\texttt{Mapper}_{LDM}$ to project $\texttt{[IMG]}$ tokens to the input space of first-stage finetuned LDM with $L$ learnable query embeddings $(q_1, ..., q_L) \in \mathbb{R}^{L \times d}$, where $L$ is the maximum input sequence length of the LDM, similar to BLIP-2 Q-Former (Li et al., 2023b). The training objective is to minimize the distance between $\texttt{Mapper}$'s output $\texttt{Gen Emb}$ and the augmented conditional text representations of LDM, i.e., $\texttt{Fuse Emb}$ introduced in section 3.2, formulated as:

$$\mathcal{L}_{\text{align}} = ||\texttt{Mapper}_{LDM}(h_{\texttt{[IMG}_{1:R}\texttt{]}}, q_1, ...q_L) - c_i||_2^2 \tag{6}$$

where $h_{\texttt{[IMG}_{1:R}\texttt{]}}$ denotes the last hidden states of LLM's $\texttt{[IMG]}$ tokens. Suppose we can get access to the original text without reference $S_i'$. Then, $c_i$ is the augmented text embedding of caption $S_i'$ encoded by the first-stage model's text and visual encoder. For instance, if $S_i$ is *"They are talking to each other"*, then $S_i'$ would be "Fred and Wilma are talking to each other." This non-referential text, augmented with character visual features, assists LLM in efficiently disambiguating references using casual language modeling.

In addition to $\mathcal{L}_{align}$, we leverage pixel-level loss $\mathcal{L}_{img}$ to facilitate semantic alignment and visual consistency. More specifically, the $\texttt{Gen emb}$ is used as a condition input for the frozen Char-LDM. The Unet $\epsilon_\theta$ of pretrained Char-LDM is used to calculate the $\mathcal{L}_{img}$ to provide pixel-level supervision defined as follow:

$$\mathcal{L}_{img} = \mathbb{E}_{\epsilon \in \mathcal{N}_{(0,1)}, t} \left[ ||\epsilon - \epsilon_\theta(z_t, t, \texttt{Mapper}_{LDM}(h_{\texttt{[IMG}_{1:R}\texttt{]}}, q_1, ...q_L))||_2^2 \right] \tag{7}$$

**Inference.** During the inference process, the model sequentially visualizes stories grounded on text descriptions. It begins by processing the text description of the initial frame $S_1$. Focusing exclusively on frame generation, we constrain the LLM to generate only $R$ specific $\texttt{[IMG]}$ tokens and then feed these token embeddings into the first-stage Char-LDM, resulting in the generation of the first frame $I_1^{gen}$. Subsequently, the LLM takes a contextual history that includes the text description of the first frame $S_1$, the generated first frame $I_1^{gen}$, and the text description of the second frame $S_2$ as input. This process is repeated to visualize the entire story progressively.

| Models | Ref text | Char-Acc (↑) | Char-F1 (↑) | BG-Acc (↑) | BG-F1 (↑) | FID (↓) | BLEU4 (↑) | CIDEr (↑) |
|---|---|---|---|---|---|---|---|---|
| StoryDALL-E† (Maharana et al., 2022) | | 69.49 | 83.35 | 48.46 | 55.24 | 44.24 | 0.4666 | 1.4473 |
| LDM (Rombach et al., 2022) | | 85.66 | 93.41 | 54.85 | 62.04 | 32.05 | 0.5230 | 1.8048 |
| Story-LDM (Rahman et al., 2023) | × | 82.43 | 91.86 | 55.3 | 61.58 | 36.29 | 0.4656 | 1.4335 |
| Char-LDM (Ours) | | **90.36** | **95.76** | **58.36** | **63.92** | **21.13** | **0.5260** | **1.8361** |
| StoryDALL-E† (Maharana et al., 2022) | | 61.83 | 78.36 | 48.10 | 54.92 | 44.66 | 0.4460 | 1.3373 |
| LDM (Rombach et al., 2022) | | 75.37 | 87.54 | 52.57 | 58.41 | 32.36 | 0.4911 | 1.5103 |
| Story-LDM (Rahman et al., 2023) | ✓ | 77.23 | 88.26 | 54.97 | 60.99 | 36.34 | 0.4585 | 1.4004 |
| StoryGPT-V (Ours) | | **88.45** | **94.94** | **56.45** | **62.09** | **21.71** | **0.5037** | **1.6718** |

Table 1: Main experiments on FlintStonesSV (Gupta et al., 2018). The top portion is evaluated on the dataset w/o extended referential text. The bottom half displays the results on the extended dataset with co-reference. †StoryDALL-E (Maharana et al., 2022) takes the source frame as additional input.

# 4 EXPERIMENTS

## 4.1 EXPERIMENTAL SETUPS

**Datasets.** Our experiments are conducted using two story visualization datasets: FlintstonesSV (Gupta et al., 2018) and PororoSV (Li et al., 2019). FlintstonesSV (Gupta et al., 2018) contains 20132-training, 2071-validation, and 2309-test stories with 7 main characters and 323 backgrounds, while PororoSV (Li et al., 2019) consists of 10,191 training samples, 2,334 for validation, and 2,208 for testing with 9 main characters. We follow (Rahman et al., 2023) to extend the datasets with referential text, by replacing the character names with references, i.e., he, she, or they, wherever applicable. Please refer to the supplementary for details.

**Evaluation metrics.** To measure the accuracy of the characters and background in the generated stories, we consider the following evaluation metrics the same as previous story visualization literature (Maharana & Bansal, 2021; Maharana et al., 2022; Rahman et al., 2023): Following (Maharana & Bansal, 2021), we finetune Inception-v3 to measure the classification accuracy and F1-score of characters (Char-Acc, Char-F1) and background (BG-Acc, BG-F1) respectively. In addition, we consider the Frechet Inception Distance (FID) score, which compares the distribution between feature vectors from real and generated images for quality assessment.

When assessing text-image alignment, the CLIP (Radford et al., 2021b) score falls short in reliability since it cannot capture fine-grained details. Therefore we choose the powerful captioning model BLIP2 (Li et al., 2023b) as the evaluation model and fine-tune it on the corresponding datasets. We then employ it as a captioner to predict 5 captions for generated images and 5 captions for ground truth images as a comparison to report the average BLEU4 (Papineni et al., 2002) and CIDEr (Vedantam et al., 2015) score to assess text-image alignment.

**Comparison Approaches.** We compare our model with state-of-the-art approaches: VLCStoryGAN (Maharana & Bansal, 2021), StoryDALL-E (Maharana et al., 2022), LDM (Rombach et al., 2022) and Story-LDM (Rahman et al., 2023). Following previous research (Li et al., 2019; Rahman et al., 2023), we use 4 consecutive frames for evaluation. For StoryDALL-E (Maharana et al., 2022), which takes both story descriptions and the initial frame as input, we use the first frame of a 5-frame story and evaluate using the generated 4 frames. We finetune vanilla Stable Diffusion (LDM) on FlintStonesSV (Gupta et al., 2018) and PororoSV (Li et al., 2019) as a baseline. Since Story-LDM (Rahman et al., 2023) does not provide pre-trained checkpoint or cleaned training code, we initiate training from pre-trained LDM[2].

**Implementation Details.** For the first stage training, we freeze CLIP (Radford et al., 2021b) text encoder and fine-tune the remaining modules for 25k steps with a learning rate 1e-5 and batch size of 32 on original non-referential text. To enhance inference time robustness and flexibility, we adopt a training strategy that includes 10% unconditional training, i.e., classifier-free guidance (Ho & Salimans, 2022), 10% text-only training, and 80% character-augmented fuse training (section 3.2). We use the original loss of latent diffusion and $\mathcal{L}_{reg}$ (eq. (2)) loss for the first stage training.

---

[2]https://ommer-lab.com/files/latent-diffusion/nitro/txt2img-f8-large

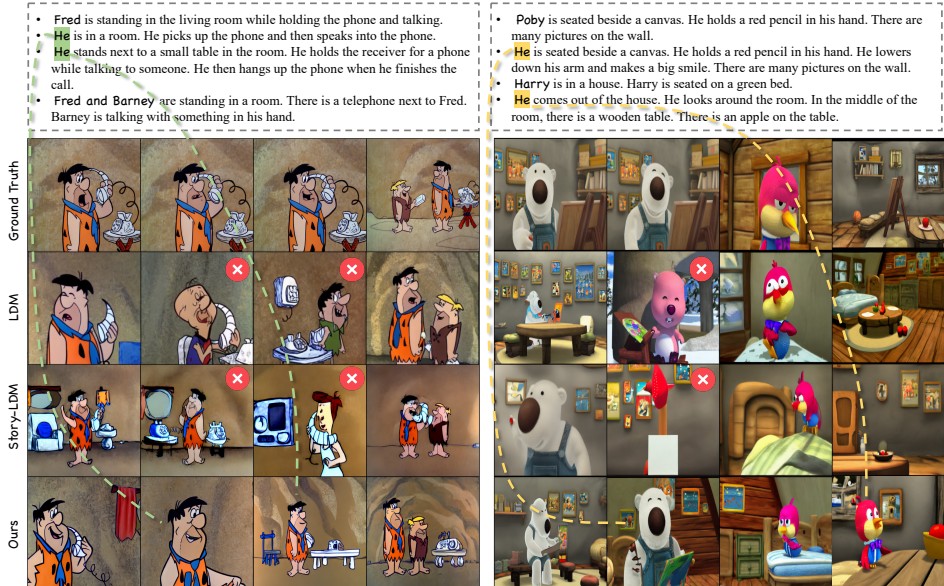

Figure 3: Qualitative comparison on FlintStonesSV (Gupta et al., 2018) (left) and PororoSV (Li et al., 2019) (right) with co-reference descriptions.

For the second stage training, we use OPT-6.7B[3] model as the LLM backbone. Please refer to the result with Llama2 (Touvron et al., 2023) as the LLM backbone in the supplementary. To expedite the second stage alignment training, we first pre-compute non-referential fused embeddings residing in the input space of the first-stage Char-LDM. We map visual features into $m = 4$ token embeddings as LLM input, set the max sequence length as 160 and the number of additional [IMG] tokens represents for LLM's visual output as $R = 8$, batch size as 64 training for 20k steps. Please refer to the supplementary for more details. We use $\mathcal{L}_{gen}$ (eq. (4)), $\mathcal{L}_{align}$ (eq. (6)) and $\mathcal{L}_{img}$ (eq. (7)) losses for the second stage training.

## 4.2 VISUAL STORY GENERATION

**Quantitative Results.** *(i) Generation with original descriptions.* The upper half of table 1 shows the comparison results on original FlintStonesSV (Gupta et al., 2018) without referential text descriptions. Our first-stage Char-LDM exhibits superior performance in generating accurate characters (Char-Acc, Char-F1) and background scenes (BG-Acc, BG-F1), achieving high fidelity (FID), and exhibiting better alignment with given text descriptions (BLEU4 (Papineni et al., 2002), CIDEr (Vedantam et al., 2015)). *(ii) Generation with co-referenced descriptions.* table 1 (bottom) and table 2 show the results on extended FlintStonesSV (Gupta et al., 2018) and PororoSV (Li et al., 2019) with co-referential text descriptions (Rahman et al., 2023) respectively. By harnessing the merit of reasoning and comprehension abilities of LLM, our model substantially boosts performance in reference resolution compared to baselines, while maintaining a strong text-image alignment grounded in the provided text descriptions.

| Models | Char-Acc (↑) | Char-F1 (↑) | FID (↓) | BLEU4 (↑) | CIDEr (↑) |
|---|---|---|---|---|---|
| StoryDALL-E[†] (Maharana et al., 2022) | 21.03 | 50.56 | 40.39 | 0.2295 | 0.3666 |
| LDM (Rombach et al., 2022) | 27.81 | 57.02 | 28.98 | 0.2560 | 0.5122 |
| Story-LDM (Rahman et al., 2023) | 29.14 | 57.56 | 26.64 | 0.2420 | 0.4581 |
| StoryGPT-V (Ours) | **36.06** | **62.70** | **19.56** | 0.2586 | **0.5279** |

Table 2: Performance comparison on PororoSV (Li et al., 2019) with co-referenced descriptions. [†]StoryDALL-E (Maharana et al., 2022) takes the source frame as additional input.

---

[3]https://huggingface.co/facebook/opt-6.7b

**Qualitative Results.** fig. 3 demonstrates qualitative comparison on FlintStonesSV (Gupta et al., 2018) and PororoSV (Li et al., 2019) with co-reference descriptions. LDM (Rombach et al., 2022) could generate high-quality images but struggles to produce correct characters in the presence of reference in the captions. Story-LDM (Rahman et al., 2023), despite incorporating an attention-memory module to handle context, struggles to produce accurate characters in some frames. In comparison, our model excels at generating frames with pleasing visuals, accurate characters, and maintaining temporal consistency in the background scenes.

**Human Evaluation.** In addition, we use Mechanical Turk to assess the quality of 100 stories produced by our methods or Story-LDM (Rahman et al., 2023) on FlintStonesSV (Gupta et al., 2018). Given a pair of stories generated by Story-LDM (Rahman et al., 2023) and our model, MTurkers are asked to decide which generated four-frame story is better w.r.t visual quality, text-image alignment, character accuracy, and temporal consistency. Each pair is evaluated by 3 unique workers. In fig. 6(a), our model demonstrates significantly better story visualization quality with accurate and temporally coherent synthesis.

### 4.3 ABLATION STUDIES

**First stage ablation.** We conducted an ablation study for the first stage and presented results in table 3. *w/o* $\mathcal{L}_{reg}$ indicates that we disabled the $\mathcal{L}_{reg}$ loss (eq. (2)), i.e., the model underwent training without the influence of segmentation masks to direct the cross-attention maps. *w/o augmented text* signifies that the model's conditional input during its training phase was the standard CLIP (Radford et al., 2021b) text embedding, rather than the fused embedding incorporating the character's visual attributes as discussed in section 3.2. *freeze vis* denotes the visual encoder remained frozen during training. Unless specified, the last two layers of the visual encoder are made adjustable. The final two rows employ our default training strategy and the only distinction lies in the inference phase. *Default (w/o img)* takes vanilla CLIP (Radford et al., 2021b) text embedding as input condition, whereas *Default (w/ img)* employs the fused embedding. As indicated by table 3, integrating character visual features during training significantly enhances the generation performance and the additional cross-attention control propels the model to achieve its peak on accurate character generation. Note that the FID score of *Default (w/ img)* is slightly higher than *Default (w/o img)*. This is because, during inference, the reference images for corresponding characters in *Default (w/ img)* are obtained online, introducing a slight deviation from the original distribution.

| Models | Char-Acc (↑) | Char-F1 (↑) | BG-Acc (↑) | BG-F1 (↑) | FID (↓) |
|---|---|---|---|---|---|
| w/o $\mathcal{L}_{reg}$ | 88.86 | 95.21 | 55.50 | 60.77 | 23.51 |
| w/o augmented text | 87.45 | 94.70 | 57.67 | 63.04 | 21.27 |
| freeze vis | 88.67 | 95.14 | 56.58 | 62.46 | 22.01 |
| Our stage1 (w/o img) | 89.73 | 95.56 | 56.18 | 62.85 | **20.96** |
| Our stage1 (w/ img) | **90.36** | **95.76** | **58.36** | **63.92** | 21.13 |

Table 3: Ablation study for the first stage finetuning LDM with cross-attention control.

**Second stage ablation.** As shown in table 4, we conducted an ablation study on (i) whether to align with the text embedding ($\text{Emb}_{text}$) or the fused embedding ($\text{Emb}_{fuse}$) of our first stage model; (ii) whether the model's input consists of a sequence of captions (Caption-) or utilizes interleaved training with both images and captions (Interleave-) (eq. (5)). Experimental results shown in table 4 indicate that image-text interleave training can significantly enhance performance. It is intuitive that taking both images and corresponding captions as input provides a more profound comprehension of the characters and their interactions within the image than when provided with sole captions. This, in turn, amplifies its generative capabilities. In addition, the $\mathcal{L}_{img}$ loss introduces pixel-level supervision, further improving visual consistency by propagating pixel-level generation to the `[IMG]` token representation of LLM.

### 4.4 ANALYSIS

We further investigate the impact of first-stage finetuning with cross-attention control by visualizing averaged cross-attention maps in U-Net latent pixel space and interpolating them to match the size of

| Models | Char-Acc (↑) | Char-F1 (↑) | BG-Acc (↑) | BG-F1 (↑) | FID (↓) |
|---|---|---|---|---|---|
| Caption-Emb$_{text}$ | 69.70 | 83.37 | 52.67 | 58.78 | 21.32 |
| Caption-Emb$_{fuse}$ | 71.77 | 84.81 | 52.57 | 58.04 | 24.79 |
| Interleave-Emb$_{text}$ | 86.10 | 93.46 | 54.92 | 60.15 | 21.30 |
| Interleave-Emb$_{fuse}$ w/o $\mathcal{L}_{img}$ | 87.96 | 94.17 | 56.01 | 61.07 | 21.71 |
| Interleave-Emb$_{fuse}$ w $\mathcal{L}_{img}$ (default) | 88.45 | 94.94 | 56.45 | 62.09 | 21.71 |

Table 4: Second stage training strategy ablation. Input only caption or interleaved text and image. The output of LLM is aligned with our Char-LDM text embedding (Emb$_{text}$) or character-augmented fused embedding (Emb$_{fuse}$).

the generated images. As illustrated in fig. 5, vanilla LDM (top) finetune on FlintStonesSV (Gupta et al., 2018) w/o $\mathcal{L}_{reg}$ (section 3.2) struggles to accurately focus on the corresponding characters for character tokens. Our model (bottom), which incorporates cross-attention guidance, is able to precisely direct attention to generated characters given corresponding character tokens.

## 4.5 PROPERTIES

Our model could generate *longer stories* featuring *accurate characters*, at a *faster speed* and with *lower computational consumption*. Our architecture allows our model to retain an extensive context requiring minimal computational resources by efficiently mapping visual features into tokens instead of operating in pixel space. fig. 6 shows the comparison between our model and Story-LDM (Rahman et al., 2023) w.r.t GPU memory consumption and inference speed for longer-frames story generation. Our model is capable of producing sequences exceeding 50 frames with low memory usage, whereas Story-LDM (Rahman et al., 2023) encounters GPU memory limitations (80G A100) when generating 42 frames. This is because Story-LDM (Rahman et al., 2023) requires the retention of the entire context, e.g., $n$ frames in latent pixel space ($n \times h \times w \times d$), whereas our model processes visual features as four token embedding ($n \times 4 \times d$) with the same dimensions as the text tokens in LLM. table 5 compares the accuracy of generated characters and FID score for long story visualizations between our model and Story-LDM (Rahman et al., 2023). The performance of Story-LDM (Rahman et al., 2023) significantly decreases when generating longer stories and reaches the memory limit before 50 frames. In contrast, by utilizing the capacity of LLM to retain extensive context, our model upholds accurate character consistency in visualizing lengthy narratives with co-referential text descriptions.

| Models | Metric | 4 | 10 | 20 | 40 | 50 |
|---|---|---|---|---|---|---|
| Story-LDM (Rahman et al., 2023) | Char-Acc (↑) | 77.23 | 74.84 | 69.01 | 63.40 | N/A |
| | FID (↓) | 36.34 | 48.92 | 53.32 | 60.33 | N/A |
| StoryGPT-V (Ours) | Char-Acc (↑) | 85.44 | 84.63 | 82.86 | 81.04 | 80.92 |
| | FID (↓) | 27.08 | 38.91 | 42.60 | 48.37 | 61.23 |

Table 5: Longer-frames story visualization comparison on FlintStonesSV (Gupta et al., 2018) with referential text. Story-LDM reaches maximum GPU capacity when generating 50 frames.

## 5 CONCLUSION

In this paper, we aim at high-quality and consistent character synthesis for story visualization grounded on co-referential text descriptions. To accomplish this, we utilize the strengths of the LDM for generating high-quality images, combined with the reasoning capability of LLM to comprehend extended contexts, resolve ambiguities, and ensure semantic consistency in the generation process. We first finetune LDM by guiding the cross-attention map of LDM with character segmentation masks, which improves the accuracy and faithfulness of character generation. Next, we facilitate a mapping from the output of LLM to align with the input space of the first stage LDM, thus allowing Multi-modal LLM to both process and produce images. This process leverages the LLM's logical reasoning to clarify ambiguous references and its capacity to retain contextual information. Our model reports superior quantitative results and consistently generates characters with remarkable quality.

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

## A  MULTI-MODAL STORY GENERATION

Owing to StoryGPT-V design leveraging the advanced capabilities of Large Language Models (LLMs), it exhibits a unique proficiency in that it can extend visual stories. StoryGPT-V is not merely limited to visualizing stories based on provided textual descriptions. Unlike existing models, it also possesses the innovative capacity to extend these narratives through continuous text generation. Concurrently, it progressively synthesizes images that align with the newly generated text segments.

Figure 4 presents an example of a multi-modal story generation. Initially, the first four frames are created according to the text descriptions from the FlintstonesSV (Gupta et al., 2018) dataset (gray part). Subsequently, the model proceeds to write the description for the next frame (blue part), taking into account the captions provided earlier, and then creates a frame based on this new description (blue part). This method is employed iteratively to generate successive text descriptions and their corresponding frames.

Our model represents a notable advancement in story visualization, being the first of its kind to consistently produce both high-quality images and coherent narrative descriptions. This innovation opens avenues for AI-assisted technologies to accelerate visual storytelling creation experiences by exploring various visualized plot extensions as the story builds.

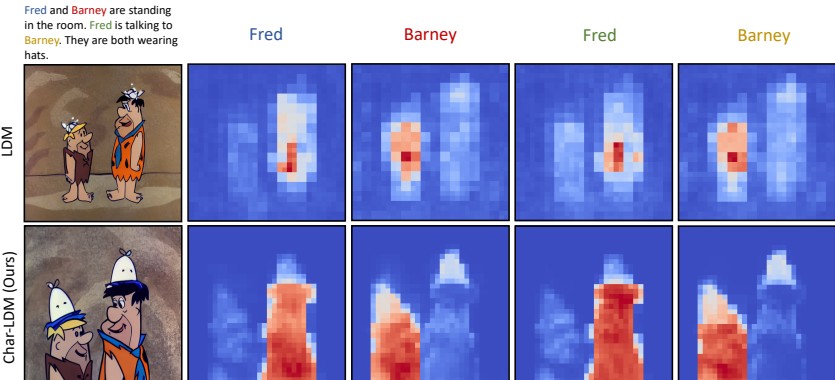

| Fred is looking over the food on the table in the dinning room. | Wilma is speaking to Fred in the dinning room. | Fred is in the kitchen. He talks while lokking at a giant pile on the table. | He is in the dinning room. He puts his hands on his hips as he talks. | Wilma says (excitedly) Oh boy, look at all the food! | Wilma looks at Fred in disblief. | Wilma rolls her eys and mutters under her breath. | Fred looks at Wilma with a mischievous grin on his face. |

Figure 4: **Our model StoryGPT-V extending stories in both language and vision:** Gray part is the text descriptions from datasets. Blue part corresponds to the model-generated frames and the continued written stories based on the previous captions.

Figure 5: Visualization of cross attention maps of corresponding character tokens.

## B    ABLATION STUDIES

### B.1    EFFECT OF FIRST-STAGE DESIGN.

In Table 6 lower half, we conducted an ablation study on how the stage-1 design contributes to the final performance. In the first line, the stage-2 LLM is aligned with vanilla LDM fine-tuned on FlintstonesSV (Gupta et al., 2018). The second line aligns the LLM output with our Char-LDM's text embedding ($\text{Emb}_{text}$), while the last line aligns with character-augmented fused embedding ($\text{Emb}_{fuse}$) of our Char-LDM. The first two lines align to the same text embedding encoded by the CLIP (Radford et al., 2021b) text encoder, however, our Char-LDM enhanced with cross-attention control ($\mathcal{L}_{reg}$) produces more precise characters. Different from $\text{Emb}_{text}$, the last line is aligned with $\text{Emb}_{fuse}$, which is augmented with characters' visual features. This visual guidance helps LLM to interpret references more effectively by linking "he, she, they" to the previous language and image context.

| Models | Aligning space | Char-Acc (↑) | Char-F1 (↑) | BG-Acc (↑) | BG-F1 (↑) | FID (↓) |
|---|---|---|---|---|---|---|
| Vanilla LDM (Rombach et al., 2022) | × | 75.37 | 87.54 | 52.57 | 58.41 | 32.36 |
| Our Stage-2 | Vanilla LDM $\text{Emb}_{text}$ | 84.06 | 92.54 | 53.18 | 58.29 | 22.94 |
| | Char-LDM $\text{Emb}_{text}$ | 86.10 | 93.46 | 54.92 | 60.15 | 21.30 |
| | Char-LDM $\text{Emb}_{fuse}$ (default) | 88.45 | 94.94 | 56.45 | 62.09 | 21.71 |

Table 6: The output of our stage-2 model (OPT) is aligned with conditional input of vanilla LDM (Rombach et al., 2022) (finetuned on FlintstonesSV (Gupta et al., 2018)), our Char-LDM text embedding ($\text{Emb}_{text}$) or character-augmented fused embedding ($\text{Emb}_{fuse}$).

### B.2   NUMBER OF `[IMG]` TOKENS

We further examined the impact of the number of added `[IMG]` tokens. As indicated in Table 7, aligning with the fused embedding and setting $R = 8$ yields the best performance.

| Models | R | Char-Acc ($\uparrow$) | Char-F1 ($\uparrow$) | BG-Acc ($\uparrow$) | BG-F1 ($\uparrow$) | FID ($\downarrow$) |
|---|---|---|---|---|---|---|
| $\text{Emb}_{text}$ | 4 | 82.14 | 90.18 | 54.28 | 59.58 | 21.33 |
| $\text{Emb}_{text}$ | 8 | 86.10 | 93.46 | 54.92 | 60.15 | 21.30 |
| $\text{Emb}_{text}$ | 16 | 83.77 | 91.07 | 54.08 | 60.21 | 21.58 |
| $\text{Emb}_{fuse}$ | 4 | 86.23 | 93.43 | 54.57 | 59.61 | 21.97 |
| $\text{Emb}_{fuse}$ | 8 | 88.45 | 94.94 | 56.45 | 62.09 | 21.71 |
| $\text{Emb}_{fuse}$ | 16 | 85.35 | 91.96 | 52.93 | 58.86 | 23.73 |

Table 7: StoryGPT-V Ablations: Impact of $R$, the number of added `[IMG]` tokens. $\text{Emb}_{text}$: the output of LLM (OPT) is aligned with text embedding extracted from the text encoder; $\text{Emb}_{fuse}$: aligned with fused embedding $\text{Emb}_{fuse}$ of first stage model.

### B.3   DIFFERENT LLMS (OPT VS LLAMA2)

| Models | # Params | Char-Acc ($\uparrow$) | Char-F1 ($\uparrow$) | BG-Acc ($\uparrow$) | BG-F1 ($\uparrow$) | FID ($\downarrow$) | BLEU4 ($\uparrow$) | CIDEr ($\uparrow$) |
|---|---|---|---|---|---|---|---|---|
| OPT (Zhang et al., 2022) | 6.7b | 88.45 | 94.94 | 56.45 | 62.09 | 21.71 | 0.5037 | 1.6718 |
| Llama2 (Touvron et al., 2023) | 7b | 89.08 | 95.07 | 57.29 | 62.62 | 21.56 | 0.5169 | 1.7516 |

Table 8: Performance on FlintstonesSV (Gupta et al., 2018) dataset with referential text using different LLMs.

Our primary contribution lies in leveraging Large Language Models (LLMs) for reference resolution for consistent story visualization. In our work, we experimented with OPT-6.7b[4] and Llama2-7b-chat[5] models. It's important to note that the utilization of Llama2 was specifically to demonstrate its additional capability for multi-modal generation. The ablation study of different LLMs was not the main focus of our research.

Our findings, as illustrated in Table 8, indicate only a slight improvement when changing from OPT (Zhang et al., 2022) to Llama2 (Touvron et al., 2023). This marginal difference is attributed to the evaluation metric's emphasis on image-generation capabilities, which assesses whether the model's visual output aligns well with first-stage Char-LDM's conditional input space.

## C   EVALUATION

### C.1   TEXT-IMAGE ALIGNMENT.

CLIP (Radford et al., 2021b) is trained on large-scale image-caption pairs to align visual and semantic space. However, a domain gap exists between pre-train data and the story visualization benchmark. Therefore, we finetune CLIP (Radford et al., 2021b) on the story visualization datasets. However, we found it still hard to capture fine-grained semantics, either text-image (T-I) similarity or image-image similarity (I-I), i.e., the similarity between visual features of generated images and corresponding ground truth images.

Upon this observation, we choose the powerful captioning model BLIP2 (Li et al., 2023b) as the evaluation model. We finetune BLIP2 on FlintstonesSV (Gupta et al., 2018) and PororoSV (Li et al., 2019), respectively, and employ it as an image captioner for generated visual stories. We avoided direct comparisons to bridge the gap between BLIP2's predictions and the actual ground truth captions. Instead, we used the fine-tuned BLIP2 to generate five captions for each ground truth image and one caption for each generated image. and report average BLEU4 (Papineni et al., 2002) or CIDEr (Vedantam et al., 2015) score based on these comparisons.

---

[4]https://huggingface.co/facebook/opt-6.7b
[5]https://huggingface.co/meta-llama/Llama-2-7b-chat

| Models | CLIP (T-I) (↑) | CLIP (I-I) (↑) | BLEU4 (↑) | CIDEr (↑) |
|---|---|---|---|---|
| StoryDALL-E (Maharana et al., 2022) | 0.4417 | 0.8112 | 0.4460 | 1.3373 |
| LDM (Rombach et al., 2022) | 0.5007 | 0.8786 | 0.4911 | 1.5103 |
| Story-LDM (Rahman et al., 2023) | 0.4979 | 0.8795 | 0.4585 | 1.4004 |
| StoryGPT-V (Ours OPT) | 0.5106 | 0.889 | 0.5070 | 1.6607 |

Table 9: Text-image alignment score for FlintstonesSV (Gupta et al., 2018) with referential text descriptions in terms of CLIP (Radford et al., 2021b) similarity, BLEU4 (Papineni et al., 2002) and CIDEr (Vedantam et al., 2015).

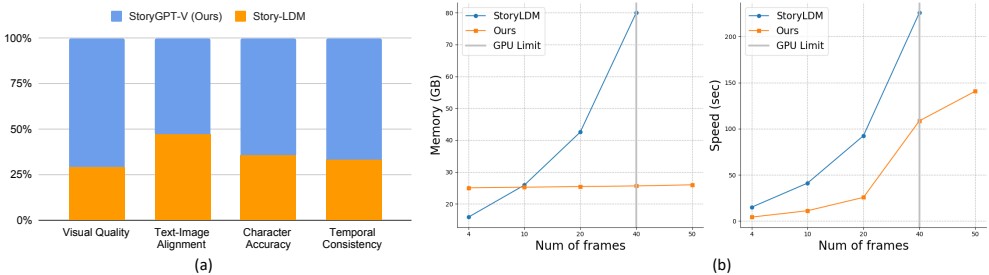

Figure 6: (a) Human evaluation results on FlintStonesSV (Gupta et al., 2018) w.r.t visual quality, text-image alignment, character accuracy and temporal consistency. (b) Compare inference speed and GPU memory consumption between our method and Story-LDM (Rahman et al., 2023). Story-LDM encounters the 80GB GPU limit when generating sequences exceeding 40 frames.

## C.2 HUMAN EVALUATION.

We use Mechanical Turk to assess the quality of 100 stories produced by our methods or Story-LDM (Rahman et al., 2023) on FlintStonesSV (Gupta et al., 2018). Given a pair of stories generated by Story-LDM (Rahman et al., 2023) and our model, people are asked to decide which generated four-frame story is better w.r.t visual quality, text-image alignment, character accuracy and temporal consistency. Each pair is evaluated by 3 unique workers. The human study interface is illustrated in Figure 7.

## C.3 OPEN DOMAIN EVALUATION

We mainly focus on closed-domain story visualization and character synthesis with ambiguous references. VIST is a story visualization data but lacks consistent visual stories as it relies on people crafting stories for 5 selected photos from a Flickr album. And it doesn't contain character/background labels for a comprehensive evaluation in the setting of consistent story visualization like (Gupta et al., 2018). We report CLIP image similarity and LPIPS score following (Koh et al., 2023a) in Table 10.

| Models | CLIP-I (↑) | LPIPS (↓) |
|---|---|---|
| LDM (Rombach et al., 2022) | 0.598 | 0.704 |
| Story-LDM (Rahman et al., 2023) | 0.504 | 0.715 |
| StoryGPT-V (Ours) | 0.613 | 0.692 |

Table 10: Results on VIST (Huang et al., 2016) dataset.

# D IMPLEMENTATION DETAILS

## D.1 DATA PREPARATION

FlintstonesSV (Gupta et al., 2018) provides the bounding box location of each character in the image. We fed the bounding boxes into SAM (Kirillov et al., 2023) to obtain the segmentation map of corresponding characters. This offline supervision from SAM is efficiently obtained without the need for manual labeling efforts.

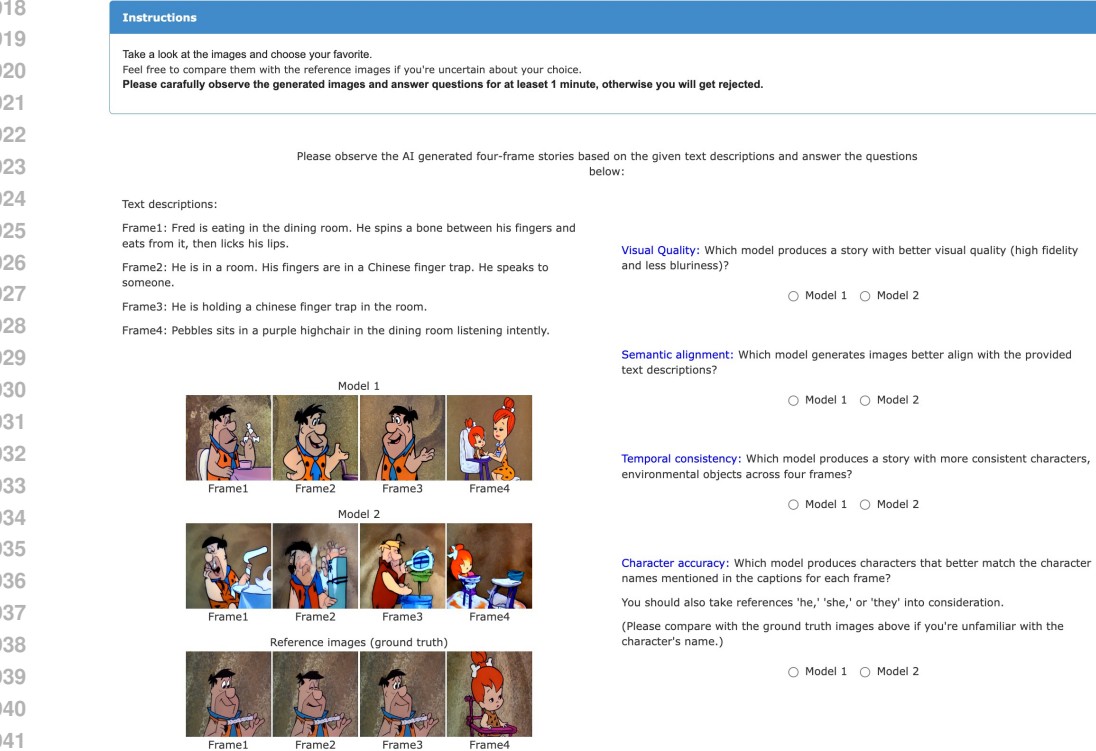

Figure 7: Human study interface.

## D.2 EXTENDING DATASET WITH REFERENTIAL TEXT

We follow Story-LDM (Rahman et al., 2023) to extend the datasets with referential text by replacing the character names with references, i.e., he, she, or they, wherever applicable as shown in Algorithm 1. The statistics before and after the referentail extension are shown in Table 11. Please refer to Story-LDM (Rahman et al., 2023) implementation[6] for more details on how the referential dataset is extended.

| Dataset | # Ref (avg.) | # Chars | # Backgrounds |
|---|---|---|---|
| FlintstonesSV (Gupta et al., 2018) | 3.58 | 7 | 323 |
| Extended FlintstonesSV | 4.61 | 7 | 323 |
| PororoSV (Li et al., 2019) | 1.01 | 9 | None |
| Extended PororoSV | 1.16 | 9 | None |

Table 11: Dataset statistics of FlintstonesSV (Gupta et al., 2018) and PororoSV (Li et al., 2019)

## D.3 FIRST STAGE TRAINING

We built upon pre-trained Stable Diffusion (Rombach et al., 2022) v1-5[7] and use CLIP (Radford et al., 2021a) ViT-L to extract characters' visual features. We freeze the CLIP text encoder and fine-tune the remaining modules for 25,000 steps with a learning rate of 1e-5 and batch size of 32. The first stage utilizes solely the original text description without extended referential text. To enhance inference time robustness and flexibility, with or without reference images, we adopt a training strategy that includes $10\%$ unconditional training, i.e., classifier-free guidance (Ho & Salimans, 2022), $10\%$ text-only training, and $80\%$ augmented text training, which integrates visual features of characters with their corresponding token embeddings.

---

[6]https://github.com/ubc-vision/Make-A-Story/blob/main/ldm/data
[7]https://huggingface.co/runwayml/stable-diffusion-v1-5

## D.4 Second stage training

We use OPT-6.7B[8] model as the LLM backbone in all experiments in the main paper. To expedite the second stage alignment training, we first pre-compute non-referential fused embeddings residing in the input space of the first-stage Char-LDM. We map visual features into $m = 4$ token embeddings as LLM input, set the max sequence length as 160 and the number of additional `[IMG]` tokens as $R = 8$, batch size as 64 training for 20k steps. Llama2 is only trained for the experiments highlighted in the supplementary materials, demonstrating its capability for multi-modal generation and the ablation of different LLMs. The training configuration is almost the same as OPT, except for batch size 32. All experiments are executed on a single A100 GPU.

Please refer to all the details at the source code.

---

**Algorithm 1** Character Replacement Algorithm

---

**Definitions:**
$i$: index for frames, ranging from 1 to $N$
$S_i$: text description of frame $i$
$\mathcal{C}_i$: a set contains immediate character(s) in the current frame
**for** $i \in \{1, 2, \ldots, N\}$ **do**
  **if** $i = 1$ **then**
    $\mathcal{C}_i \leftarrow$ immediate character of $S_i$
  **else**
    **if** $\mathcal{C}_i \subseteq \mathcal{C}_{i-1}$ **then**
      **if** $\text{length}(\mathcal{C}_i) = 1$ **then**
        Replace $\mathcal{C}_i$ in $S_i$ with "he" or "she"
      **else if** $\text{length}(c) > 1$ **then**
        Replace $\mathcal{C}_i$ in $S_i$ with "they"
      **end if**
    **end if**
    $\mathcal{C}_i \leftarrow \mathcal{C}_{i-1}$
  **end if**
**end for**

---

## E Limitations

Our method demonstrates proficiency in resolving references and ensuring consistent character and background conditions in the context provided by guiding the output of a multi-modal Large Language Model (LLM) with character-augmented semantic embedding. However, several limitations remain. The process involves feeding the previously generated frame into the LLM to produce a visual output that aligns with the Latent Diffusion Model (LDM) input conditional space. This approach guarantees semantic consistency, enabling the generation of characters and environmental objects that resemble their originals. Nonetheless, there are minor discrepancies in detail. This is because the visual output from the Large Language Model (LLM) is aligned with the semantic embedding space rather than the pixel space, which hinders the complete reconstruction of all elements in the input image. However, the current most powerful multi-modal LLM, i.e., DALL-E 3 (OpenAI, 2023), could not solve this exact appearance replication in the multi-round image generation task (Figure 8), indicating an area ripe for further exploration and research.

## F Qualitative Results

We provide more generated samples on FlintstonesSV (Gupta et al., 2018) and PororoSV (Li et al., 2019) with referential text as Figure 9-18 show.

---

[8]https://huggingface.co/facebook/opt-6.7b

- Fred is standing in the living room while holding the phone and talking.
- He is in a room. He picks up the phone and then speaks into the phone.
- He stands next to a small table in the room. He holds the receiver for a phone while talking to someone. He then hangs up the phone when he finishes the call.
- Fred and Barney are standing in a room. There is a telephone next to Fred. Barney is talking with something in his hand.

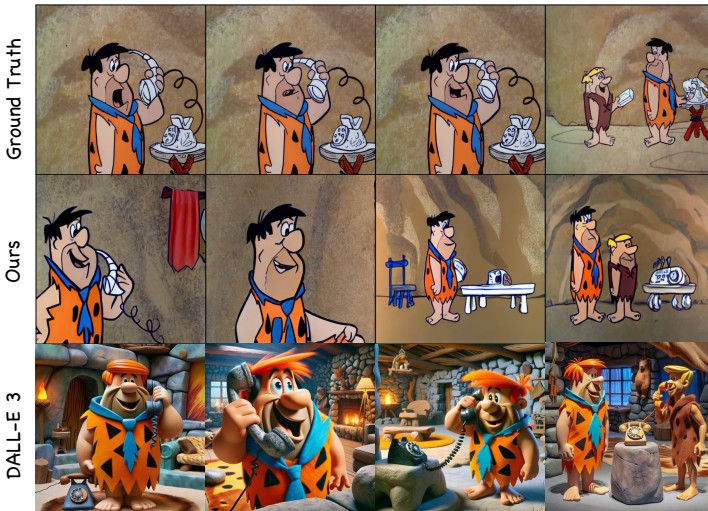

Figure 8: DALL-E 3 (OpenAI, 2023) zero-shot inference on FlintstonesSV (Gupta et al., 2018) dataset.

- Barney is in the dining room at the table. He is holding a stack of papers and talking.
- He stands in the room, laughing at a newspaper.
- He opens a box while holding papers in a room. Then he hold the papers with both hands and laughs.
- Betty is sitting on a chair in the living room.

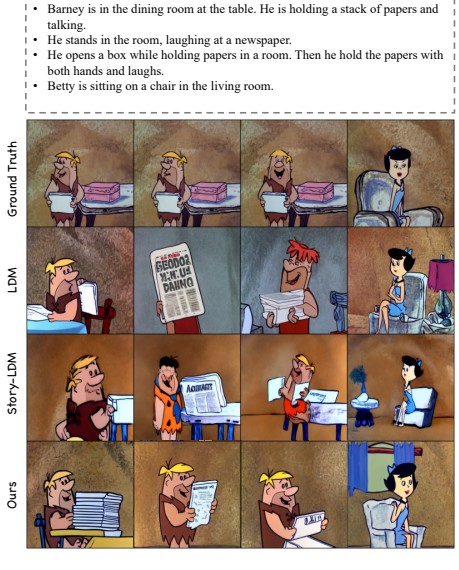

Figure 9: Qualitative comparison on FlintstonesSV (Gupta et al., 2018) with co-reference descriptions.

- Wilma is in the room. She at first has her eyes closed and then opens them.
- Fred is standing in the living room while talking.
- He is in the living room with arms stretched out.
- He is standing in the doorway of the living room, talking to someone off screen right.

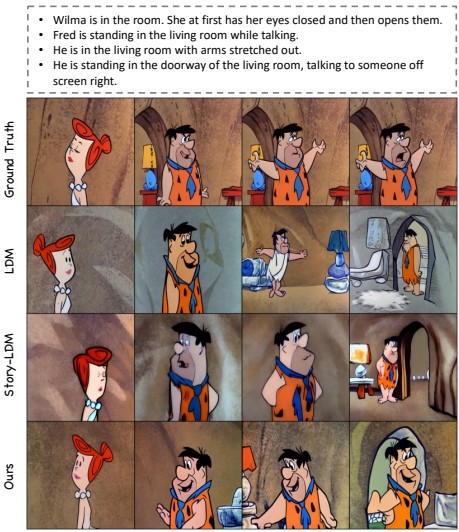

Figure 10: Qualitative comparison on FlintstonesSV (Gupta et al., 2018) with co-reference descriptions.

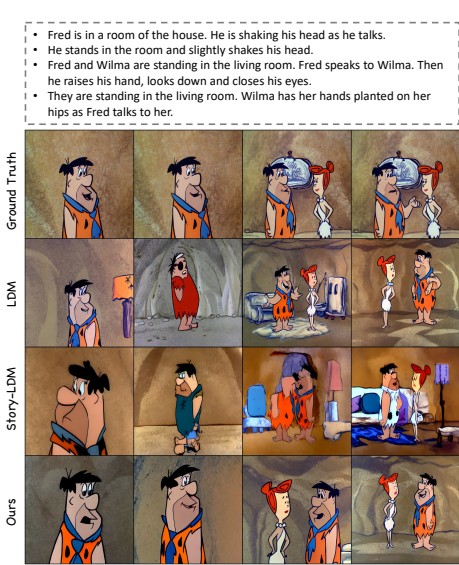

Figure 11: Qualitative comparison on Flint-stonesSV (Gupta et al., 2018) with co-reference descriptions.

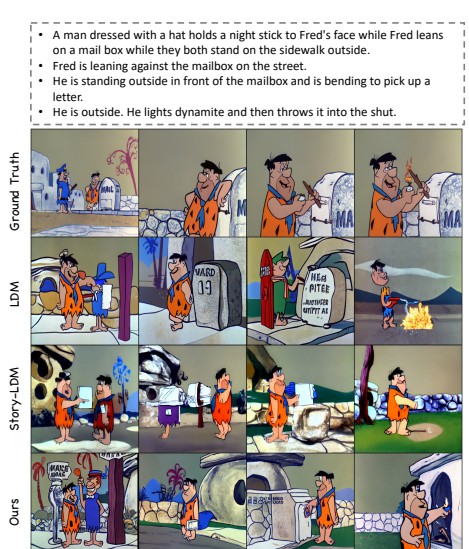

Figure 12: Qualitative comparison on Flint-stonesSV (Gupta et al., 2018) with co-reference descriptions.

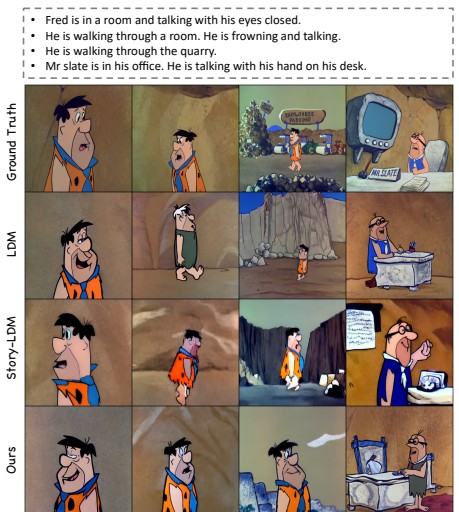

Figure 13: Qualitative comparison on Flint-stonesSV (Gupta et al., 2018) with co-reference descriptions.

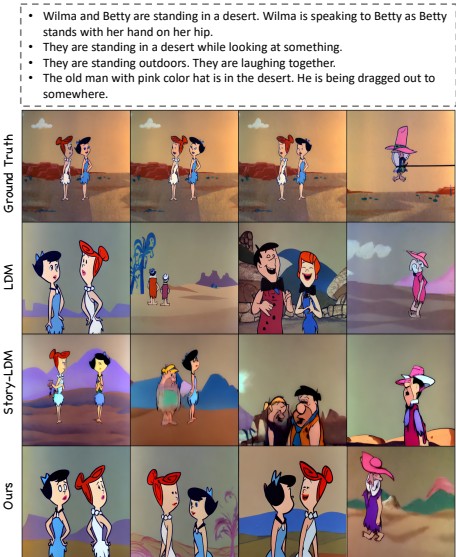

Figure 14: Qualitative comparison on Flint-stonesSV (Gupta et al., 2018) with co-reference descriptions.

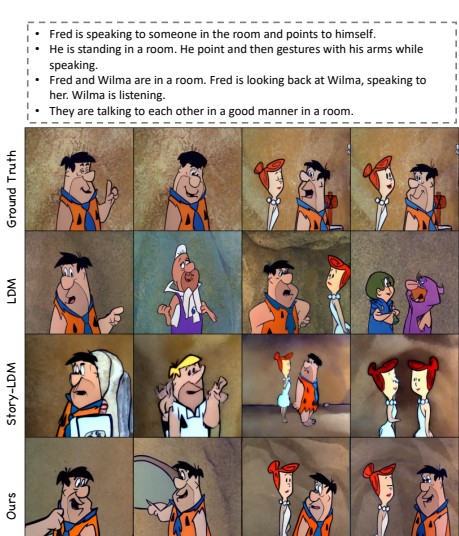

Figure 15: Qualitative comparison on Flint-stonesSV (Gupta et al., 2018) with co-reference descriptions.

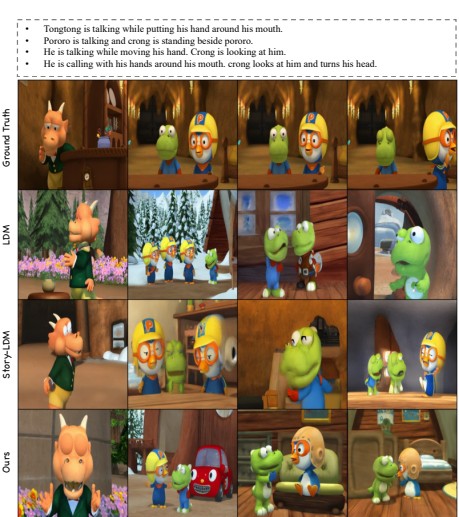

Figure 16: Qualitative comparison on Poro-roSV (Li et al., 2019) with co-reference descriptions.

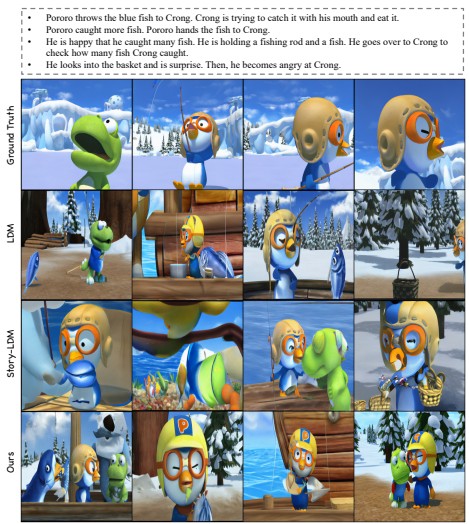

Figure 17: Qualitative comparison on Poro-roSV (Li et al., 2019) with co-reference descriptions.

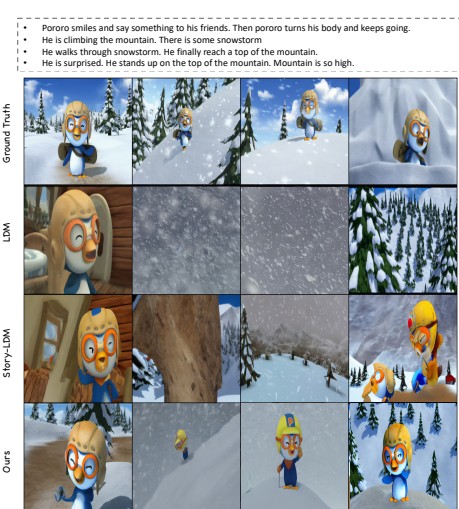

Figure 18: Qualitative comparison on Poro-roSV (Li et al., 2019) with co-reference descriptions.

