# OpenReview forum: "StoryGPT-V: Large Language Models as Consistent Story Visualizers"
_ICLR.cc/2025/Conference — ICLR 2025 Conference Withdrawn Submission_

### Official Review · Reviewer_dKW1 · 2024-11-03

**Soundness:** 2
**Presentation:** 2
**Contribution:** 2
**Rating:** 3
**Confidence:** 3

**Summary:**

This paper tackles the story visualization task and propose a two-stage approach. First, the authors propose Char-LDM, a finetuned LDM by guiding the cross-attention map of LDM with character segmentation mask. Second, they utilize Multi-modal LLM for retaining contexts and reference resolution by mapping from the output of LLM to align with the input space of the first stage LDM. In the experimental results, it is shown that the proposed approach outperforms baselines including Story-LDM with lower computational resources such as GPU memory  for longer image frames and inference speed.

**Strengths:**

- The proposed idea of using segmentation mask for guiding the cross-attention map of LDM makes sense, and using LLM for retaining contexts and reference resolution is interesting.
- In experimental results, it is shown that the proposed model outperforms the baselines in both quantitative and qualitative evaluations.

**Weaknesses:**

- Novelty seems somewhat incremental as using segmentation mask of characters for generating them is not new (e.g., [1]), and also for LLM to understand and generate images, it uses the similar idea from prior work such as [2].
- In the experiments, there are points to raise concerns regarding the technical soundness of them. First, it is said: "Story-LDM (Rahman et al., 2023) does not provide pre-trained checkpoint or cleaned training code" (line #368), so it was not clear how it was implemented even though it is written: "we initiate training from pre-trained LDM". Furthermore, in Table 1 (bottom), the reported Char-acc and Char-F1 (77.23 & 88.26) in the paper are different from what are reported in Story-LDM paper (69.19 & 86.59). Second, in human evaluation, only three judges were used and it seems too few.
- Writing needs to be clearer. e.g., it is not clear whether only first stage (Char-LDM) was used or both stages were used for the result in Table 1 (upper half) as it is written: "Our first-stage Char-LDM exhibits superior performance in generating accurate characters" (line #414).
- The claim of using less GPU memory is questionable. My understanding is 2nd stage needs to load LLM to memory, so it seems more expensive compared to the baseline, i.e., Story-LDM, especially for the typical scenario for story visualization of generating five image sequence. This seems also confirmed in Figure 6(b) when the number of frames to generate is less than 10.
- It lacks an important ablation study: using only first stage without second stage to see the effectiveness of LLM.

References

[1] Song et al., Character-Preserving Coherent Story Visualization, ECCV 2020

[2] Koh et al., Generating Images with Multimodal Language Models, NeurIPS 2023

**Questions:**

- Notation: what is m in line #282? Also is k the same as e in line #282-283?
- In Figure 6(b), how did you use longer stories? Did you extend existing stories with LLM?
- Please address weaknesses mentioned above.

Typo:
Line #014: visualization. Since -> visualization since

---

### Official Review · Reviewer_FoYq · 2024-11-04

**Soundness:** 1
**Presentation:** 2
**Contribution:** 2
**Rating:** 3
**Confidence:** 5

**Summary:**

This work targets to tackle the ambiguous subject reference challenge in text-to-image visual storytelling. The core idea is to leverage the comprehension and reasoning ability of Large Language Model. First, the authors train a character-aware Latent Diffusion Model. Then, the character-aware semantic embeddings from LDM and LLM output are aligned. Experiments are conducted on two story visualization benchmarks and show more consistent character generation.

**Strengths:**

The work addresses an important problem in visual storytelling for enhanced multiple character consistency, leveraging LLM's comprehension and reasoning ability.

**Weaknesses:**

1. The paper writing needs improvements. For example, (1) the sentence "Since it requires resolving pronouns (he, she, they) in
the frame descriptions, i.e., anaphora resolution, and ensuring consistent characters and background synthesis across frames." is incomplete in the abstract L14. (2) The core contribution and benefit over previous MLLM alignment approaches is unclear. Suggest to elaborate the advantage of the proposed ''implicitly deduced reference'' explicitly (L109, L139).
2. Incomprehensive related work. For instance, StoryGen (CVPR'24), StoryDiffusion (NeurIPS'24), and SeedStory ('24) are recent work towards subject consistency in story visualization. In particular, SeedStory ('24) also leverages LLM's reasoning ability and aligns text-image.
3. Incomplete experiments. It misses important comparisons to the published StoryGen (CVPR'24) and StoryDiffusion (NeurIPS'24). Suggest to include in Table 1 and 2.

**Questions:**

1. What is the benefit of the proposed multimodal alignment over previous methods? It'd make the work more solid by conducting an ablation study comparing the proposed alignment method to counterpart alignment approaches (Table 3), measuring performance on key metrics like character consistency or reference resolution accuracy.

---

### Official Review · Reviewer_WK9q · 2024-11-04

**Soundness:** 3
**Presentation:** 2
**Contribution:** 2
**Rating:** 6
**Confidence:** 4

**Summary:**

This paper proposes StoryGPT-V, a novel framework that integrates LLMs with LDMs to enhance the ability to resolve pronouns and consistent generation. It employs a two-stage training process: first, it trains a character-aware LDM to take character embeddings as input; second, it aligns the output of LLM with the input space of LDM. Results show that LLMs significantly enhance the model's ability to understand pronouns, and StoryGPT-V shows good capability to generate consistent stories.

**Strengths:**

- Pronoun resolution is an interesting topic in storytelling. Introducing LLMs to address this problem is a very reasonable approach. The cross-attention map visualized on the anonymous webpage demonstrates that it can effectively assist LDMs in associating pronouns with characters.

- This paper conducts comprehensive ablation experiments to demonstrate the effectiveness of the proposed training strategy in the two-stage training process. Additionally, it shows better competitiveness across various metrics compared to the methods listed on the FlintStonesSV Benchmark.

**Weaknesses:**

My main concern is that the paper does not sufficiently discuss the latest methods.

- Before this paper, there were various storytelling papers(e.g., TaleCrafter[1] in SIGGRAPH Asia 2023, AR-LDM[2] in CVPR 2024) that integrated LLMs into diffusion models, and some works(ConsiStory[3] in TOG 2024) also manipulate attention modules for consistent story generation. A concurrent work, SEED-Story[4], also shares a similar idea that utilizes an LLM for multimodal story generation, and the diffusion model serves as a visual decoder. These works are highly relevant and deserve discussion.
- Furthermore, some works have also conducted experiments on FlintStonesSV, such as AR-LDM and TaleCrafter. Adding comparisons with these works could potentially enhance the paper's credibility, if possible.

[1] Talecrafter: Interactive story visualization with multiple characters
[2] Synthesizing coherent story with auto-regressive latent diffusion models
[3] Training-free consistent text-to-image generation
[4] SEED-Story: Multimodal Long Story Generation with Large Language Model

**Questions:**

Please check out the Weaknesses part.

---

### Official Review · Reviewer_jH7x · 2024-11-11

**Soundness:** 3
**Presentation:** 3
**Contribution:** 2
**Rating:** 5
**Confidence:** 4

**Summary:**

This paper focuses on the task of story visualization and proposes combining the advantages of LLMs and LDMs to generate a consistent story from a given long narrative. To achieve this, the proposed model consists of two stages: the first stage introduces visual features into the text embedding to produce a fused embedding, which is then used in the second stage to generate results. Experimental results demonstrate that the proposed method effectively generates high-quality, consistent outputs.

**Strengths:**

1. The paper proposes combining two powerful models, LLMs and LDMs, for the story visualization task, and experiments show promising results from this combination.

2. Experiments indicate that fusing visual features into text embeddings to refine an LDM can improve the quality of the output results.

3. Autoregressive generation, which continuously utilizes visual features from the previous frame, helps maintain consistency across different frames in a story and enables the model to generate longer narratives.

**Weaknesses:**

1. The strong performance of the proposed method primarily relies on the powerful capabilities of both LLMs and LDMs. Although the authors have introduced a way to combine both models for the story visualization task, the technical novelty may be limited.

2. I have concerns about the diversity of the proposed model. In the first stage, the authors incorporate visual features into text embeddings and use segmentation masks to supervise LDM training. In this setup, the output results of the refined LDMs for a given story may be constrained by the infused visual features. Based on this, I am curious about the diversity of outputs when the same story is provided as input.

3. The paper emphasizes that the proposed method can effectively address consistency issues, such as resolving pronouns in frame descriptions; however, the authors provide only qualitative results for this claim. Including quantitative results could better support the effectiveness of the approach.

4. It may be beneficial to include recent works such as StoryGen and StoryDiffusion, as well as more complex datasets like the StorySalon dataset.

StoryGen, StorySalon dataset: Chang Liu, Haoning Wu, Yujie Zhong, Xiaoyun Zhang, Yanfeng Wang, and Weidi Xie. Intelli- gent grimm-open-ended visual storytelling via latent diffusion models. In Proceedings of the IEEE/CVF Conference on Computer Vision and Pattern Recognition, pp. 6190–6200, 2024.

Story-Diffusion: Yupeng Zhou, Daquan Zhou, Ming-Ming Cheng, Jiashi Feng, and Qibin Hou. Storydiffusion: Con- sistent self-attention for long-range image and video generation. NeurIPS 2024, 2024.

**Questions:**

See above weaknesses.

---

### Note · Authors · 2024-11-13

I have read and agree with the venue's withdrawal policy on behalf of myself and my co-authors.